# Are Dual-Purpose Chickens Twice as Good? Measuring Performance and Animal Welfare throughout the Fattening Period

**DOI:** 10.3390/ani10111980

**Published:** 2020-10-28

**Authors:** Inga Tiemann, Sonja Hillemacher, Margit Wittmann

**Affiliations:** 1Institute of Animal Science, University of Bonn, Endenicher Allee 15, 53115 Bonn, Germany; shill@uni-bonn.de; 2Institute of Agricultural Engineering, University of Bonn, Nussallee 5, 53115 Bonn, Germany; 3Faculty of Agriculture, South Westphalia University of Applied Sciences, Lübecker Ring 2, 59494 Soest, Germany; wittmann.margareta@fh-swf.de

**Keywords:** dual-purpose chicken, male layer chicks, growth performance, animal welfare, sampling methods, European production efficiency factor, welfare protocol

## Abstract

**Simple Summary:**

Dual-purpose chickens and native chicken breeds have gained public interest and their agricultural use has increased. Furthermore, the fattening of so-called “brother cockerels” of laying hens has been periodically discussed, with an increasing number of questions raised about their high feed consumption and low fleshiness and, therefore, low sustainability. This study carefully examines chickens—native, hybrid dual-purpose, and layer lines—to assess their performance in meat production, in addition to animal-based welfare indicators and sampling methods. Compared to modern broilers, tested alternatives gain less meat, consume more food, and need a longer fattening period. However, advantages may still exist in terms of animal welfare and with regard to ethically justifiable animal production in the future. Of the three alternatives tested, dual-purpose hybrids may be suitable for niche production, but further research is needed on economically sound feed management and adapted breeding strategies for dual-purpose chickens.

**Abstract:**

Chickens are the world’s most widely used farm animal and have a significant genetic diversity. In the current study, we investigated three strains for their suitability as dual-purpose chickens, with a focus on the fattening ability and welfare of the cockerels: 1. layer cockerels (Lohmann Brown, LB, *n* = 714); 2. cockerels of a dual-purpose hybrid (Lohmann Dual, LD, *n* = 844); and 3. cockerels of a native breed (Rhinelander, RL, *n* = 458). Chicks were raised under identical conditions and marked individually to compare focus and random sampling methods for weighing birds weekly. Because chicks of dual-purpose origins are usually raised mixed-sex, cockerels and pullets were weighed and observed together until sexes the were identifiable at week 10 of their life. During the 10th to 20th week of life, investigations were continued on 100 cockerels per genotype. Key figures for growth performance, such as feed conversion ratio (FCR) and European production efficiency factor (EPEF), were also calculated at weekly intervals. LD cockerels showed considerable growth performance (*p* < 0.001 compared to LB, RL, 2 kg at 9 weeks), whereas LB reached a live weight of 2 kg at 13 weeks and RL at 15 weeks of age. Genotype-dependent differences were also evident, with favorable FCR and EPEF for LD, intermediate for LB, and unfavorable for RL (all *p* < 0.001). The results of the FCR and EPEF suggest that cockerels should be slaughtered around week 8 of life, although only the carcass of the LD might be marketable. Thus, the optimal time of slaughter based on production parameters such as FCR and EPEF is different from the time when the animal reaches a marketable 2 kg live weight. Animal-based welfare indicators revealed that the RL are not adapted to production environments, including those that are extensive. Further research aimed at adapted feed management, including better FCR, and animals adapted to the respective production environments is necessary to improve alternative poultry production in the future.

## 1. Introduction

Each year, more than 45 million one-day-old male layer chicks are killed in Germany [1], despite global ethical concerns [2,3]. Currently, chicks are sexed immediately after hatching, and the female layers are sold to farms rearing the chicks for egg production, whilst the male chicks are killed due to a lack of demand [4]. Because these male layer hybrids cannot compete with broilers raised for meat production in terms of feed efficiency or weight gain, their economics are under discussion [5,6]. Under increasing pressure from politicians and the public, and a pending ban on killing one-day-old male layer hybrids, alternatives, including so-called dual-purpose chickens, are currently being developed [6,7,8].

The use of these dual-purpose chickens appears to be a retrograde move because they combine growing males and laying females in a single genotype, as was the case prior to 1950, when layers were specialized as high-performing hens and non-fattening males, and broilers in which both sexes are able to be fattened. Although the performance of the genetics for egg and meat production of dual-purpose hybrids is low, these chickens may perform better than native chickens or brother cockerels from layer lines [6,9]. On the contrary, native chicken breeds are better adapted to their typical extensive husbandry, whereas hybrid lines are mainly produced intensively. In addition, cockerels of layer hybrid lines can be fattened. Koenig et al. [10,11] found a body weight of 650 g of layer hybrid cockerels after a fattening period of 49 days, whereas Mueller et al. [9] referred to a fattening period of nine weeks with a lower food efficiency and inferior meat quality. The majority of studies collect growth data based on randomly weighted chickens [12]. In contrast, studies on animal-based welfare indicators are often collected from focal animals [13]. This study addresses the methodological question of which sampling methods lead to more reliable data, e.g., whether the monitoring of focal animals leads to results that are closer to the mean value of the total population as measured using random animals. This might be of special interest because the variability of the data associated with the mean value, i.e., the homogeneity of the herd, is expected to be higher in high-performance than in recently developed lines, such as the Lohmann Dual, or breeds without breeding management, such as Rhinelander. 

Present challenges in poultry production also include the increasing public interest in overall animal welfare, including physiology and animal behavior. Chickens of native breeds are believed to show a better adaptation to local housing conditions, which is a desirable breeding goal in poultry [14] and has also been reported for cattle in terms of climate adaptation [15,16]. This adaptation to the production environment is, with physical health, the key factor for welfare of farm animals, including poultry [17]. To measure the welfare of broilers and laying hens, the Welfare Quality^®^ assessment protocol for poultry (broilers, laying hens) was developed, which contributes quantitative and qualitative data to the question of welfare on farms [18]. In addition to the evaluation of management- and health-related indicators, the animal-based measurement of appropriate behavior is one part of the assessment protocol. For broilers and laying hens, the Qualitative Behavior Assessment (QBA) evaluates the animals’ body language to uncover the emotional state of the animals.

The management of rearing dual-purpose chickens is challenging because it is based on unsexed groups until the sexes are recognizable at about 10 weeks. Thus, the management of dual-purpose chickens initially follows that of broilers, in which both sexes are also fattened. After the 10th week, however, the hens are reared further so that they can be used later as laying hens. The lack of sexing of day-old chicks is also due to the fact that this is a niche production and, therefore, professional, personnel structures, such as those in the hatcheries of laying hens, are not available.

In addition, the fattening period of layer-type and dual-purpose cockerels is significantly prolonged compared to that of broiler lines. This prolonged fattening period requires a prolonged view of the animal’s welfare. The Welfare Quality^®^ assessment protocol and its chapter on broiler welfare, however, is usually applied at a single farm visit and might be repeated for different flocks at the same location. The development of the animals in a given generation on the farm is usually not covered by the protocol. This study covers weekly welfare assessments of the chickens until slaughter to ensure that both their development and welfare throughout the fattening period are reliably assessed. 

This study, therefore, compares Lohmann Brown (LB, layer hybrid), Lohmann Dual (LD, dual-purpose hybrid), and the native dual-purpose breed Rhinelander (RL) chickens regarding their (1) growth and corresponding performance data, including feed efficiency, (2) performance data of focal vs. randomized birds measured, and (3) health and behavioral development of welfare indicators, particularly the development of behavioral expressions (body language) referring to the interaction with conspecifics and the environment. The decision for or against chickens of a specific genotype for production purposes is based on basic growth performance parameters, such as body weight and feed conversion ratio (FCR), subsumed in the European production efficiency factor (EPEF). Due to the special rearing concept of unsexed groups containing pullets and cockerels until their 10th week of life, these parameters are applicable to both sexes. For the final fattening period, only cockerels are kept, usually until day 80. Although focal sampling of these animals should lead to reduced variabilities, mean growth performances should be identical. We hypothesize that commercial dual-purpose chickens show an advantage in weight gain, but that the native dual-purpose breed is more adapted to extensive fattening in terms of animal welfare as assessed through the Qualitative Behavior Assessment [19]. 

## 2. Materials and Methods

In this study, chickens were reared and fattened at the Campus Frankenforst of the Faculty of Agriculture, University of Bonn (Königswinter, Germany). Hatching of Lohmann Dual (LD) and Rhinelander (RL) chicks took place at the Poultry Research Center, Rhein-Kreis Neuss, Germany, whereas Lohmann Brown (LB) chicks were obtained from a commercial hatchery (Geflügelzuchtbetriebe Gudendorf-Ankum GmbH & Co., Ankum, Germany). In total, 844 LD, 714 LB, and 458 RL chicks were transported to the rearing facilities of Frankenforst, kept unsexed but separated per genotype from week 0 to 10. After 10 weeks, all cockerels, except for 100 birds per genotype, were slaughtered, and all hens changed stables for laying purposes.

Each chick was marked individually using a wing band. Housing was a conventional floor system with access to a free-range area beginning at 13 weeks of life. All three pens, one per genotype, had an area of 33.82 m^2^ (7.6 × 4.45 m) with three windows (11% daylight in relation to the floor) and a free-range area of 210 m^2^. Chicks were raised according to a standard light program based on a 16L:8D schedule. The stables were bedded with wood shavings (Allspan^®^ classic, Allspan Spanverarbeitung GmbH, Karlsruhe, Germany) and cleaned or newly bedded on a regular basis. For the first two weeks, the stable area for the chicks was restricted by cardboard to approximately 20 m^2^ to keep the chicks next to the heat lamps. A manual ventilation system was switched on as required. Additionally, two wooden frames with three perches per frame were placed in each stable. Visual inspection, including visual health checks, took place at least twice per day, and animal losses and lost wing bands were documented. Chickens followed a vaccination plan issued by the veterinarian in charge of the flock (including vaccination against Marek’s disease, Newcastle disease, infectious bronchitis, laryngotracheitis, coccidiosis, and Gumboro). All stables were equipped with feed and water dispensers, giving the animals enough space at the dispensers to comply with the current regulatory requirements (0.66 cm/1000 g live weight [20]). Chicks were fed according to the recommendations for conventional fattening of broilers on a three-phase feeding management because there is no specific recommendation for dual-purpose chickens. According to the farmers’ feedback and due to the study design with the goal of a practical implication, dual-purpose chickens were raised unsexed until all cockerels were slaughtered at around week 10. Then, the hens’ feeding plan returned to that of a layer grower. For the first two weeks, the animals were fed 2 mm starter pellets (Landkornstarter; 21.5% crude protein, 0.6% methionine, 0.9% calcium, 0.6% phosphorus, 12.40 MJ ME/kg, coccidiostat), followed by a grower diet (Landkornmast; 21.0% crude protein, 0.5% methionine, 0.8% calcium, 0.6% phosphorus, 12.40 MJ ME/kg, coccidiostat) until one week before slaughtering, when they were fed the finisher diet (Landkornendmast; Deutsche Tiernahrung Cremer GmbH & Co KG, Düsseldorf, Germany; 20.0% crude protein, 0.5% methionine, 0.8% calcium, 0.5% phosphorus, 12.40 MJ ME/kg). Feed was weighed once per week to calculate the feed conversion ratio (FCR) corrected by the rate of mortality (losses per day). Animals had ad libitum access to feed, water, and grit at all times. Manipulable materials such as straw or apples on chains were offered additionally.

Performance measurements included the parameters growth, daily weight gain, mortality, feed consumption, and calculations based on these parameters, the FCR, and the European production efficiency factor (EPEF).

During the fattening period, live weight was measured on a weekly basis to calculate daily weight gain for at least 40 randomly selected (RL, 40 out of *n* = 458; LD, 80 out of *n* = 844; and LB, 80 out of *n* = 714) and 40 focal chickens (RL 40, LD/LB 80), which were marked with additional colored leg bands. All focal and random chickens were kept within one group per genotype in separate pens. The focal group was sampled each week in the same procedure as that of the random group, which was selected for weighing by chance. To ensure randomization of the group, animals were sequentially collected from all areas of the pen and wing tagged to ensure that they were not already weighed and did not belong to the focus group. Weighing took place on day 1 (within 24 h of hatching) and every seven days thereafter until day 141. Sexes were identifiable at an age of 10 weeks and allocated retrospectively to the individual weighed chickens. The male–female ratio was 1:1.5 in LD, 1:1.4 in LB, and 1:1.35 in RL. In total, 3536 weights were clearly assignable to either sex. Of these weights, 947 were measured on focal cockerels, whereas 890 were collected from randomly selected cockerels. After week 10, the remaining 100 cockerels per genotype were investigated with the same procedures until slaughtered at an age of 20 weeks to be able to take the weight development after maturity into account. Growth rate was calculated using (average live weight – weight at hatching) 7 days^−1^. Animal losses were monitored twice per day and resulted in a mortality rate per genotype by calculating the percentage of losses per week, accumulated for each week of life.

Production parameters, such as FCR as a marker for feed efficiency, and the EPEF, which integrates growth, mortality, and feed efficiency, were calculated as follows:FCR = feed expense [kg bird^−1^]/total weight gain [kg bird^−1^].(1)

Feed expense was measured weekly by adding the feed used minus the weekly back weighing. To compare production performance, the EPEF was calculated as:EPEF = (survival rate [100 − mortality] × live weight [kg bird^−1^]/age [days] × FCR) × 100.(2)

These data were converted into cumulative feed consumption per animal, FCR, and EPEF. Additionally, weighing results of focal animals vs. randomly sampled animals were compared statistically.

Welfare measurements included the measurement of morphological indicators and the Qualitative Behavior Assessment (QBA).

Within the QBA, a total of 22 different behaviors and interactions with other animals or the environment are surveyed (compare Appendix A; “inquisitive” was not analyzed due to missing data). The QBA was the first action on each test day (day of life 1 to 141, every seven days) to ensure that the impressions were as undisturbed as possible; only then were further data, such as weight, collected. Animals were observed within their familiar environment in the pen in the morning by the same familiar person, who was in the pen on a regular basis at least once per week. At the beginning of a testing session, two different observation points per pen were selected, which allowed the visual registration of all animals in the pen. After entering the pen and arriving at the observation point, the animals were first left undisturbed for 5 min until they returned to their previous behavior. The flock was observed in situ for 10 min (in total 20 min/genotype) per observation point. After the observation period, the occurrence of each behavior of the list of terms (pp. 30–31, [19]) was evaluated using a visual analogue scale. Each scale ranged from minimum (left hand side, resulting in lower scores) to maximum (right hand side, resulting in higher scores), with a length of 125 mm, which the observer marked accordingly. Because both negative and positive terms regarding animal welfare were surveyed, maximum values were not, per se, better or desirable. The marked point of scale was then measured, providing a quantitative value. The aim of the QBA is to scale, from minimum to maximum on the flock level, all body language (compare 5.1A.4.4; [19]).

Following the behavioral experiments, a morphological rating of the animals was carried out, monitoring their health status. Therefore, the Welfare Quality^®^ assessment protocol for poultry [18], particularly the section relating to broilers, was applied, including plumage cleanliness (score 0–3 with increasing soiling, (pp. 22–23, [19]), hock burn (scores 0–4 with increasing skin lesions, (pp. 26–27, [19]), and foot pad dermatitis (scores 0–4 with increasing dermatitis, (p. 27, [19]). For this purpose, 20 chickens of each genotype were selected randomly within their group each week and examined inside their home pen with regard to these animal-based morphological welfare indicators. Until week 10, unsexed birds were selected and examined, whereas only cocks were inspected from week 10 onwards. Using illustrated scales from the Welfare Quality^®^ assessment protocol for poultry [18], the individual variables were assigned to the appropriate categories.

Graphical presentation was performed using the program SigmaPlot 14.0 (Systat Software Inc., Chicago, IL, USA). The program SPSS^®^ Statistics 26 (IBM Corporation, Armonk, NY, USA) was used for statistical evaluation. Mean values and standard errors are given in the tables and graphs. The significance level α was set at *p* ≤ 0.05 and indicated as *, *p* ≤ 0.01, indicated as **, and *p* ≤ 0.001 as ***. Genotype-specific differences concerning weight, daily weight gain, and mortality were analyzed using a Welch-ANOVA due to lack of homogeneity of variances, followed by Games–Howell post hoc tests. EPEF and FCR were analyzed using a univariate analysis of variance (ANOVA), followed by Dunn–Bonferroni post hoc tests for pairwise comparisons. 

Further, comparisons of body weight between focal and random groups at the genotype level were undertaken using the unpaired t-test (given homogeneity of variances) or Welch test (no homogeneity of variances).

Raw data of the QBA were analyzed using the procedure given by the Welfare Quality^®^ assessment protocol for poultry [18]. First, terms were multiplied by given weights (see Welfare Quality^®^ assessment protocol). For each week of life, an index was calculated as given in 5.2.1.12 in the Welfare Quality^®^ assessment protocol. The score is limited to the range of 0 to 100. The impact of the genotype on the overall QBA score was analyzed using a Kruskal–Wallis test due to a lack of homogeneity of variances. In addition, to analyze if the course of the terms over time differs between strains, raw data of each QBA term were logarithmized in EXCEL (Microsoft Office 2016, Redmond, WA, US) to calculate the area under the curve (AUC) using: x = (t_j_ + 1 − t_j_) × (y_j_ + y_j_+1)/2,(3)
where t is the time of measurement and y is the measured value. The interquartile range was calculated as:IQR = Q_0,75_ − Q_0,25_.(4)

Each AUC term was then analyzed using a univariate analysis of variance (ANOVA) with genotype as fixed factor, followed by Dunn–Bonferroni post hoc tests.

Additionally, we followed the approach of Muri et al. [21] by using principal component analysis (PCA) to examine the components that explained most of the variance in the data. For PCA, the Kaiser–Meyer–Olkin measure of sampling adequacy and Bartlett’s test of sphericity were checked and a correlation matrix with no rotation was used.

No statistical calculations were carried out for the morphological ratings (all ratings fulfilled the scale level 0).

Animals were kept according to the order on the protection of animals and the keeping of production animals [20]. In this study, any procedures that involved handling the animals followed the instructions of the assessment protocol for poultry [18], particularly the section for broilers (see 5.1A.4.4, p. 38). The Campus Frankenforst of the Faculty of Agriculture is approved as a trial farm (39600305-547/17).

## 3. Results

### 3.1. Performance Measurements

The growth rate of the cockerels, separated into focal and random groups, is shown in Figure 1. The growth rate was clearly distinguishable between LD (dual-purpose hybrid), and LB, and RL (layer hybrid and native dual-purpose), which reached weight marks later than LD and LB. The LD achieved a live weight of 2 kg at week 8 (+2 d), whereas LB and RL reached the 2 kg live weight in weeks 12 (+4 d) and 14 (+3 d), respectively (averaged weights, Appendix A). In addition to the weight at hatching, the live weights of the birds of the different strains differed from each other at every time of measurement (all *p* < 0.001). Overall, Welch-ANOVA and post hoc tests (Games–Howell) revealed that birds of the three strains had significantly different growth rates (F(2; 1987.521) = 133.001, *p* < 0.001). In more detail, LD vs. LB *p* < 0.001 and LD vs. RL *p* < 0.001, with LD showing higher weights. The comparison of LB and RL failed to show significance (*p* = 0.057).

Focal and random groups did not differ at any time of measurement in LB (T = 0.033, df = 40, *p* = 0.974). In RL, there was no overall effect between groups (T = −0.325, df = 39, *p* = 0.747), although groups differed at weeks 2 (T = −4.301, df = 19, *p* < 0.001) and 7 (T = −2.235, df = 36, *p* = 0.032). In LD, however, focal and random groups differed from each other for the majority of weeks of life (all *p* ≤ 0.007), except for the weight at hatching, with the focal cocks being heavier than the randomly chosen cocks. However, there was no genetic-specific difference (T = 0.414, df = 40, *p* = 0.681).

Daily weight gain was calculated based on the same dataset (Appendix A). Strains differed (F(2; 74.194) = 6.971, *p* = 0.002), with LD showing the highest daily weight gain, followed by LB and then RL. Daily weight gain differed between LD and LB (*p* = 0.012), and LD and RL (*p* = 0.001), but not between LB and RL (*p* = 0.433). At the end of the fattening period, around maturity at week 20, cocks of LD and RL lost rather than gained weight.

Regarding cumulative mortality, animals of LB showed the lowest cumulative mortality by the end of the fattening period (0.7%), whereas mortality in LD (6.3%) and RL (8.1%) was comparably high. Most of the losses were documented from the 1st week (LD 2.6%, LB 0.1%, RL 1.1%) until the 4th week of life (LD 3.8%, LB 0.1%, RL 3.7%). Mortality across birds differed (F(2; 26.833) = 210.873, *p* < 0.001) based on the location of the hatch (hatchery for LB, research facilities for LD and RL; LD vs. LB *p* < 0.001; LD vs. RL *p* = 0.235; LB vs. RL *p* < 0.001).

Cumulative feed consumption amounted to 4.85 kg for LD, 3.33 kg for LB, and 3.53 kg for RL at the 10th week of life, and to 15.9 kg for LD, 11.5 kg for LB, and 15.2 kg for RL at the 20th week of life (Figure 2). Overall, this factor did not show genetic specificity (F(2; 57) = 0.798, *p* = 0.455). The calculated FCR per genotype and week is shown in Table 1. Birds differed regarding their FCR (F(2, 37.27) = 11.121, *p* < 0.001), with RL showing a higher FCR than LD (*p* < 0.001) and LB (*p* = 0.001); there was no difference between LD and LB (*p* = 0.772).

**Table 1 animals-10-01980-t001:** Feed conversion ratio (FCR, feed expense ([kg bird^−1^])/total weight gain [kg bird^−1^]) as a measure of feed efficiency, given per genotype and week of life, in addition to average FCR per genotype. Measures were taken from unsexed groups until week 10 (indicated by line), when hens were removed from the flock and only 100 cocks/genotype remained.

Week of life	Lohmann Dual	Lohmann Brown	Rhinelander
1	1.34	0.83	4.84
2	1.36	1.37	3.43
3	1.53	1.58	2.84
4	1.49	1.57	2.90
5	1.52	1.74	2.81
6	1.56	1.78	2.55
7	1.70	1.94	2.57
8	1.71	2.00	2.83
9	1.84	2.14	2.70
10	1.92	2.25	2.79
11	2.06	2.43	2.94
12	2.40	2.61	3.29
13	2.58	2.88	4.13
14	2.72	3.02	4.60
15	2.92	3.32	4.63
16	3.21	3.41	4.91
17	3.33	3.65	5.45
18	3.52	3.91	5.53
19	3.82	3.96	5.82
20	3.93	4.04	6.30
Average	2.32	2.52	3.89

The EPEF was calculated for birds of each genotype in each week of life (Figure 3). Based on the dataset, the EPEF reached its highest score in LD and LB at week 8 (192 and 97.2 points, respectively) and in RL at week 10 (61.3 points). Comparisons of the birds of all three strains revealed differences according to their EPEF, overall (F(2, 33.94) = 70.813, *p* < 0.001) and pairwise (all *p* < 0.001).

**Figure 3 animals-10-01980-f003:**
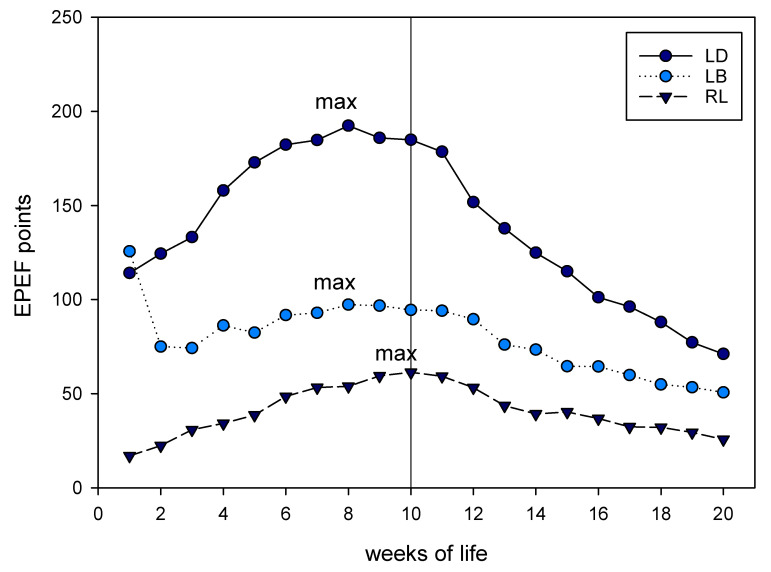
Calculated points of the European production efficiency factor (survival rate (100 – mortality) × live weight [kg bird^−1^]/age [days] × FCR) × 100) for birds of all strains. Maximum points are marked (note that for LB the local maximum score is indicated, not the global maximum). Measures were taken from unsexed groups until week 10, when hens were removed from the flock and only 100 cocks/genotype remained.

### 3.2. Welfare Measurements

Morphological assessment of the animals did not reveal any abnormalities in their general state of health. Concerning breast feather coverage and cleanliness, foot pad dermatitis, and hock burn, all scored animals were scored as 0 in each week of life. From the 15th week of life onwards, approx. 20% of the animals had slightly conspicuous foot pads (slight swelling recognizable) in the RL, but not in LD or LB. These did not correlate to other morphological scores because the latter were 0 throughout the observation period.

The Kruskal–Wallis test revealed no difference between the birds of the different strains (H(2) = 0.410, *p* = 0.815) regarding their overall QBA score. Over time, neither LB (Mdn = 81.34, IQR = 24.14), LD (Mdn = 79.97, IQR = 14.89), or RL (Mdn = 74.12; IQR = 24.12) showed lower scores in terms of their general positive emotional state. The results of the single-term analysis can be found in the Appendix A.

In the principal component analysis, the two main factors accounted for 58.57% of the overall variance (principal component 1: 47.5%; principal component 2: 11.12%; for mean values of the area under the curve per term see Appendix A; for loading terms on component 1 and 2 see Appendix A). The Kaiser–Meyer–Olkin measure of sampling adequacy was high (0.847) and resulted in a significance in the Bartlett’s test of sphericity of *p* ≤ 0.001. Figure 4 shows the loading of each expressive quality term on the two factors. Note that negative associations with the animals’ body language (agitated, unsure, scared, tense, nervous, helpless, distressed, drowsy, depressed, frustrated) had a high positive loading on component 1 and are accumulated in the upper right quadrant. All positive associations with the animals’ behavior are spaced on the left-hand side, such as positively occupied, confident, friendly, relaxed, active, energetic, and content (negative loadings on component 1). The principal component analysis based on the study’s data satisfactorily maps the aggregation of positive and negative associations to the animals’ body language.

## 4. Discussion

The study’s goal was to observe and analyze performance of chickens—commercial dual-purpose hybrid and layer hybrids, and a native dual-purpose purebred breed—during the rearing and fattening period. The initial hypotheses were that (a) commercial dual-purpose chickens show a superior growth performance, and (b) the animal-based welfare indicators of chickens of the native dual-purpose breed indicate better adaptation to extensive fattening management. Although the first hypothesis is supported by the study’s results, the second hypothesis regarding welfare measurements reveals that native breeds might be less adaptable to an environment driven by production management goals.

Lohmann hybrids are used for food production globally, whereas RL are a native breed that was first described as a farm chicken for egg and meat production by Bruno Dürigen [22]. RL has never been bred for production environments because the main breeding objective, after solely hybrid lines were developed for egg and meat production, was the outer appearance. Most breeders use their animals as food, but carcasses are not marketed. Pure breeds are typically kept in groups of one cockerel and up to five hens under free-range conditions to retain optimal fertilization and awards in exhibitions. The difference in management between the usual husbandry for RL and an extensive commercial production is not in the technical but in the social environment. Maintaining larger groups appears to lead to a reduction of welfare indicators, particularly with regard to the emotional state captured by the body language of the animals.

In this study, differences in total weight gain, daily weight gain, FCR, and EPEF were found among LD, LB, and RL. Although the (daily and total) weight gain of the LB and RL was similar, the LD weighed approximately 1200 g more than LB and RL at 20 weeks of life, as expected based on the breeding focus and previous studies. In the study of Mueller et al. [23], LD cockerels weighed 1740 g after 67 days of fattening with a conventional broiler diet, whereas cockerels of this study weighed 2565 g three days later at 70 days of fattening. In addition, LB remained below the measured weights of this study (2741 g), with weights of 2146 g at 18 weeks. These different results cannot be traced back to different feeding management approaches but might be due to changes in genetics, because LD are still a genetic line in progress. In the study of Kreuzer et al. [24], LD at 9 weeks reached comparable weights of 2127 g (compared to 2242 g here), as did LB with a weight of 1255 g (1320 g here). All studies unanimously report that performance of LB is lower than those of any other genetic line in focus, whereas for LD, some studies reported performance comparable with that of slowly growing broilers.

The higher the FCR, the higher the ecological impact and the sustainability [25]. Feeding schemes that lower the impact and increase sustainability have been studied, but have not been implemented in practice to date. Compared to the FCR of modern (1.52) or slow-growing (2.43, [9]) broilers, the FCR of dual-purpose chickens might still compete with slow-growing lines, with a range of 2.83 [23] and 1.92 for LD, whereas layer lines are less favorable, with results of 3.66 [23] and 2.25 for LB obtained in this study. The abrupt increase in feed amount around week 13 for the Rhinelander might be due to feed exploration and, therefore, wastage behavior, which may occur after exchange of litter material, as has been previously shown for LB [23]. For dual-purpose chickens, it has also been shown that the substitution of diet ingredients can be both ecologically and economically beneficial without disadvantages in production traits. A recently published study of German Vorwerk, French Bresse, and commercial White Rocks showed that soybean meal can be substituted by regionally grown legume grains, in this case Faba beans, without negative impacts on growth performance [26]. Dual-purpose alternatives are also discussed for other species, especially ruminants, with ideas to reduce the carbon footprint by raising calves on farms or using other diet compositions [27]. This extension towards system-relevant characteristics has led to an overall view which, with regard to chickens, recommends simultaneous consideration of meat and egg production [9,28]. For this reason, it should be initiated to consider the lifetime performance for both sexes in the future and, if necessary, to extend this to the parent generation (total feed efficiency; as discussed in the pig production industry [29]).

Although alternative genetic origins are being increasingly investigated, the EPEF as a comparative tool is calculated rarely. Here, we used the EPEF to determine the most reasonable slaughtering age as 8 weeks for hybrids and 10 weeks for the purebred breed. The EPEF was calculated on cockerels and pullets because both were unsexed until 10 weeks; thus, it would be of interest to calculate a more precise EPEF based on cockerels’ performance measurements only. This requires that day-old dual-purpose chickens are sexed. However, the priority management of dual-purpose chicken relies on an unsexed rearing period until about 10 weeks of life, when sexes are identifiable and cockerels have additional weeks of final fattening [6]. In addition, this “optimal” EPEF does not reflect an “optimal” marketable carcass with larger segments. The prolonged fattening period not only results in disadvantages, particularly with regard to feed efficiency, but higher slaughtering ages might also result in a higher meat quality, as shown for Chinese Da-Heng chickens [30]. Lower weight and longer rearing period could, therefore, lead to a more traditional eating experience [1] in both senses of the word: traditional breeds and traditional food. Extensive fattening, organic, or conventional free-range rearing might also be favorable if the vegetation has a positive impact on production traits, enabling the producer to offer functional food [31]. The initiation of changes within the entire value chain was investigated by Saatkamp et al. [32], originally based on higher welfare for conventional broilers. The five key factors were found to be availability, willingness to pay, actions by NGOs, initiatives of retailers, and the introduction and communication of new concepts. This outline would be transferable to dual-purpose chickens extended by the idea of the agro-ecological and sustainable conservation of native breeds [33,34,35].

Studies on growth performance usually randomly select chickens for weighing [36]. Sampling methods with focal individuals might be more common in other disciplines, such as clinical animal research [37] or ethology [38]. For weighing, approaches of random [36,39] and complete weighing of all birds [9,30] are most common, and individual marking through wing or leg bands is possible [40]. The chickens of this study were individually marked, and 40 RL, 80 LD, and 80 LB were individually traceable throughout the complete fattening period. Differences were found between the focal and random groups, with higher weights in the focal than in the random group, although only in LD. Furthermore, in LD the variances of the mean were found to be not homogenous between the focal and random groups. Two explanations are conceivable. One possibility is that the cockerels that were handled individually on a weekly basis showed less stress responses, which correlated with an increased adaptation towards the environment, resulting in increased physiological capacities of weight gain [41,42]. Because other studies of handled piglets could not find an influence on weight gain [43], an experimental bias would be obvious. The other possibility is that data reveal an underestimation of growth performance due to an experimental bias that catchers preferably weighed lighter cockerels. This would indicate that the focal group represents the population mean more appropriately, based on randomly chosen one-day-old chicks, and would be the method of choice for future studies.

Although the outcome of the study revealed new data relating to dual-purpose chickens, due to the heterogeneity of genetic origins, there are limitations to address. The low mortality found in LB compared to LD and RL might have been biased by the place of hatching. Causes for mortality can be manifold [44]. All chickens were hatched in the Poultry Research Center, whereas LB were obtained from a commercial hatchery as day-old chicks. Differences in mortality, including losses until day 10, could therefore reflect early selection processes at hatching rather than breed differences. Stocking density differed between groups increasingly due to a considerable different weight development, and particularly low animal numbers for the native breed RL because brooding eggs were extremely rare. Other influencing factors, such as the male–female ratio, were found to be equal to the tested strains and in line with that of previous studies [45]. 

With regard to the utilization of cockerels of layer lines for fattening, health and welfare are thought to be unproblematic [25], but are also widely unexamined in terms of quantifiable data throughout a growing period which was several times that of the fattening period of broilers. Studies applying standardized welfare assessment protocols to dual-purpose chickens are rarely found [46]. Using the QBA as a reliable holistic methodology to rate the animals’ emotional health is supported, for example, by studies on calves [47] and sick and recovering dairy cows [48], and has also been queried by others, e.g., based on the feasibility and reliability of the assessment protocol for pigs [49,50]. Recently, the QBA was also applied for the assessment of broiler [21] and layer welfare [51]. Here, cockerels of the LD, LB, and RL genotypes did not differ in their overall QBA scores, although the RL score was slightly lower (based on the median) than those of the commercial hybrid lines. 

We found the terms “depressed”, “helpless”, “unsure”, “scared”, “distressed”, “nervous”, and “frustrated” of the QBA to have the highest positive loading (above 0.75) on the principal component 1. Based on these terms, we were able to label the first component “mood”, as proposed by Muri et al. [21]. There were also positively connoted terms such as “confident”, “positively occupied”, “content”, and “friendly”, which had a negative load on component 1. Component 2 was described by the term “bored” (above 0.75), which had a negative load. Therefore, we concur with the label “arousal” as also proposed by Muri et al. [21].

Animal-based welfare indicators that address behavioral traits of the chicken should be taken into account in the decision process of producing a specific chicken genotype, and should be part of their future breeding schemes.

The native dual-purpose breed RL cannot compete with the modern dual-purpose hybrids in terms of daily weight gain, total weight gain, and feed efficiency. In total, the Rhinelander had the lowest daily weight gain and total weight gain, in addition to the lowest feed conversion ratio, which cannot be compensated for by a promising animal welfare assessment. Other native breeds, such as Belgian Malines and Swiss chicken (Schweizerhuhn) [9], showed performances that offered either no, or only niche in the case of French Bresse Gauloise [6], potential in mobile husbandry systems. As found in the Rhinelander, welfare problems, e.g., in terms of foot pad dermatitis, should be carefully monitored and, in addition to performance, should also be part of future breeding schemes and management guides. Rizzi et al. [52] recently showed that Italian purebred breeds had more broken and floor eggs in a free-range system, which indicates the need for breeding management in populations of native chickens [53]. An integrative part of this life cycle should also be the hen and its laying performance [28]. In general, a life cycle assessment [27] including the genotype chosen, the time of slaughtering, and the footprint left in terms of overall sustainability should be conducted to evaluate chicken performance.

## 5. Conclusions

The substantial diversity of dual-purpose chickens represents a significant opportunity for alternative poultry production. However, not all chickens are either marketable or adapted to commercial production management. For all three genetic backgrounds, this study shows the corresponding fattening performance as a basis for decisions on the use and future of alternative origins. The highest growth performance was achieved by a dual-purpose hybrid Lohmann Dual, followed by the layer hybrid Lohmann Brown and native purebred breed Rhinelander. The same evaluation was also based on the welfare indicators, which showed that the native breed is inferior to the economic hybrids. However, dual-purpose chickens require a new way of looking at performance data in particular, because rearing is based on unsexed chicken groups and is therefore completely different from commercial farming. The initial, joint rearing and the later, gender-specific fattening or further rearing of hens requires a novel management approach to meet the requirements of the animals of their environment and the keepers of their animals. Determination of the most suitable genotype will not only depend on growth characteristics but also on the welfare and behavior adapted to the offered and communicated production niche.

## Figures and Tables

**Figure 1 animals-10-01980-f001:**
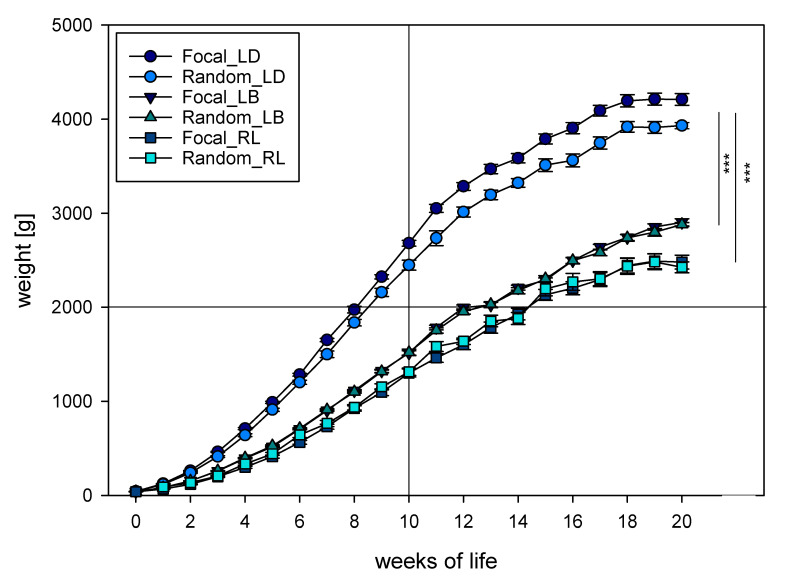
Growth rates (g) of cockerels of commercial dual-purpose hybrid Lohmann Dual (LD), commercial layer hybrid Lohmann Brown (LB), and native purebred breed Rhinelander (RL) until the 21st week of life. Shown are means ± SE of focal and random sample groups; significant differences are indicated with stars (*** for *p* ≤ 0.001). Indicated are reference lines for 10 weeks of life and 2000 g live weight.

**Figure 2 animals-10-01980-f002:**
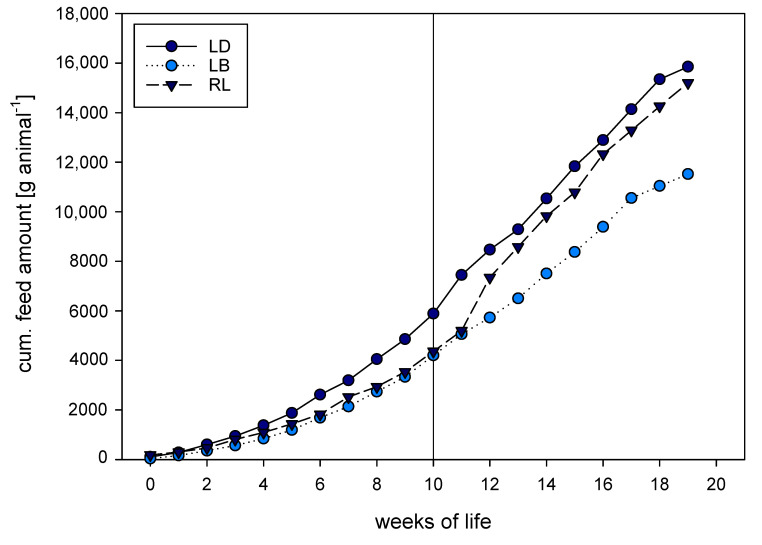
Cumulative feed amount in g per animal of commercial dual-purpose hybrid Lohmann Dual (LD), commercial layer hybrid Lohmann Brown (LB), and native purebred breed Rhinelander (RL) over the entire fattening period. Measures were taken from unsexed groups until week 10 (indicated for reference), when hens were removed from the flock and only 100 cocks/genotype remained.

**Figure 4 animals-10-01980-f004:**
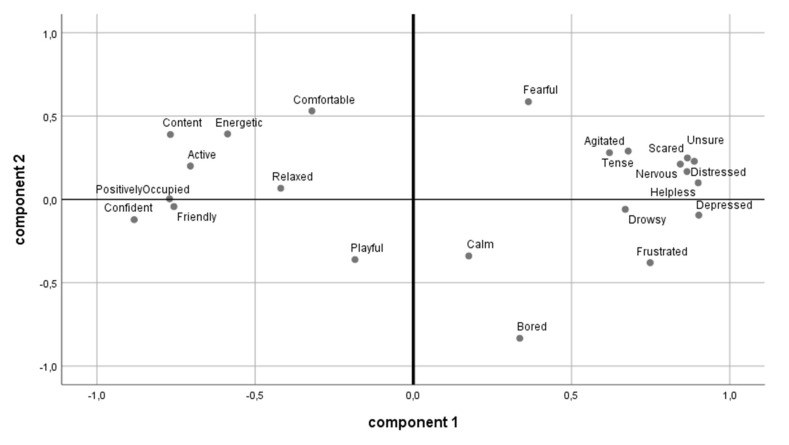
View of the principal component analysis and the loading of each term in components 1 and 2.

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
