# Peer review of "Are Dual-Purpose Chickens Twice as Good? Measuring Performance and Animal Welfare throughout the Fattening Period"

_animals, 2020, doi:10.3390/ani10111980_

Round 1

Reviewer 1 Report

The major issues with this manuscript is that the methodology is not described in enough detail and that the methodology is not tied well enough into the goals and aims of the manuscript. The authors need to refocus the paper on to specified aims and hypotheses and provide enough details to allow the readers to understand the purpose and outcomes of the paper, and to allow for reproducibility. The authors seemed to have used QBA without a thorough understanding of it or the statistical methods needed to analyze it.

Introduction

L51-52: Redundant of L49.

L53-L56: The intention of this paragraph is unclear to me.

L63-64: The five freedoms don’t specifically mention human-animal relationship, unless you mean that this is important in removing fear.

L75-79: What are the overall aims of this project? What are your hypotheses?

Would “breeds” or “strains” be more appropriate than “genotype”?

Materials and Methods

L86-87: Please provide the definitions for the acronyms again.

L115: Can you specify how you plan on evaluating “fattening performance” and “performance data” and “health and behavioral development” as specified in the project goals? In the results, you mention growth, mortality, and feed efficiency as measures of performance, but these are not explicitly outlined in the methods section.

L119: Change to “the focal group was sampled each week”

L 120: How were the birds selected at chance?

L126: Please define FCR.

L126: How was growth rate calculated? What about mortality?

L135: How were they randomly selected?

L141: Was QBA done at the same time as the other measures? Was it done weekly?

L139-L142: You have not provided enough details about how you have conducted QBA. The Table with the QBA results is Table 4. How did you conduct the QBA? Where and how were the animals observed? Who observed the animals? Did you use free choice profiling or were the observers provided a list of descriptive words and asked to rank their observations on a visual analogue scale? Additionally, the statistical analyses for QBA are quite specific and based on the methods you used to observe the animals. Within the statistical analysis section, I do not see information on how you analyzed the QBA data.

L154-155: What was the purpose of comparing the focal and random groups?

L 156: This is the only place in the manuscript where you use an abbreviation for Welfare Quality Assessments. Please spell it out fully or provide the acronym earlier in the manuscript.

L164-L165: Can you specify which test was used for each variable?

Results

L183-L186: This should be explained in the methods section.

Figure 1: Please provide the key for this figure.

L187-197: Is it possible to specify the differences on the figure itself and highlight the main points in text. This paragraph is hard to read.

L187: Change to: “the growth rate of the birds from the three genotypes”

L 188: Change “growth” to “growth rate.”  

L193: Specify “the birds of different genotypes” or something similar throughout instead of just saying “genotypes.”

Table 1: Is it necessary to discuss daily weight gain and growth rate, or are they redundant? Is there a reason these two measures are both needed?

L208: What is the significance of comparing focal and random groups?

Figure 2: Is it redundant to have feed amount and feed conversion ratio?

Figure 3: Wouldn’t the max score for LB be the one farthest to the left?

Table 4: The details of analysis are not provided in the Methodology section. The placement of AUC and Breed is confusing.

Why did you compare focal and random birds for some measures and not others?

Discussion

Please start the discussion by restating the goals and hypotheses of the research. Make sure the discussion is focused on the research aims and results that address those aims.

L280-296: See earlier comments on QBA.

L300: What do you mean by outer appearance? Do people not use these birds for meat and eggs on a smaller scale?

L302-303: One might argue intensive agriculture overburdens the birds and their natural behavior for all breeds.

L307-308: Your expectations or hypotheses should have been provided earlier in the manuscript.

Conclusions

The conclusion does not appropriately summarize the aims of the paper and the results.

L380-381: Specify male chicks.

Author Response

Dear Reviewer #1,

We thank you for your valuable comments and suggestions. In the attached file, we report any changes to the text and additional explanations. We are positive that we improved the manuscript accordingly and hope that these alterations find your support.

Yours sincerely,

the authors.

Reviewer 2 Report

The objective of this research was to assess the performance and welfare of cockerels from three genotypes (heritage strain, commercial dual-purpose strain, and commercial layer strain) during fattening. The results for this study support their conclusions and provide preliminary data about the performance of cockerels from different genotypes raised for meet production as a solution to ethical concerns regarding male-chick culling at hatch. However, the main two limitations of the experimental design of this study is that there is only one replicate per genotype and each genotype was raised in different conditions (unequal group size and density per pen and probably unequal proportion of male to female [data not provided in text] in each pen) until 10 weeks of age. I think that authors should address these limitations and its implications in the interpretation of these results in the discussion section.

Additionally, the methodology needs further information, justification, and clarification about how data was collected, especially in the case of the welfare indicators. For example, it is unclear the number of males from which the performance outcomes were collected before 10 weeks of age and what observations were collected from only cockerels vs cockerels and pullets together.

Overall, the manuscript well written and well structured. Consider the follow comments below to improve the quality of the manuscript.

LINE 25: Replace ‘cocks’ by ‘cockerels’ and throughout text.

LINE 90: Include a statement indicating that pullets and cockerels were raised together by genotype from week 0 to 10 and indicate the number of pullets and cockerels at 10 weeks of age after sexing.

LINE 91: Replace ‘stable’ by ‘pen’.

LINE 96: Include information about the light program (e.g. L:D schedule and light intensity) used in this study.

LINE 105: Poultry starter diets are commonly mash or crumble rather than pellets because chicks need small particle size for easy feed intake. Is this a typo?  

LINE: How did researchers select chickens randomly? What was the random criteria?

LINE 117-118: How many of these random and focal birds were males before 10 weeks of age?

LINE 121: Indicate the male to female ratio for each genotype in this study.

LINE 122: ‘Only data of male chickens were analyzed’. This statement can be misleading and is not correct for mortality, FCR, and EPEF data prior to 10 weeks of age. Indicate what measurements and performance outcomes refer to both sexes.

LINE 133: I suggest describing the QBA first followed by physical scoring due to chronology.

LINE 135: How many of these birds were males?

LINE 136: Why did you only assess the feather coverage in the breast area? Feather pecking and severe pecking are main welfare problems in poultry resulting in feather coverage loss in any other body area. Why did not you look at the feather coverage in the head, neck, back, and vent?

LINE 138: How many categories were considered for each indicator? Do high scores refer to poor or good health?

LINE 139 – 142: “Table 1” is missing and further information on how the QBA data was collected is needed. How many cockerels per genotype were assessed? How was the QBA done? What was the higher score for each term/emotional category? Was the QBA done in situ or recorded on camera? How old were the focal birds when the QBA was done? For how long the QBA observation lasted? Did you collect repeated measures at different ages? At what time of the day was the QBA performed?

LINE 181: I find confusing that data collected from cockerels is presented together with the performance outcomes that were collected from males and females together (mortality, feed intake, FCR, and EPEF). Particularly when the objective of this study is to describe the performance and welfare of cockerels from three genotypes. I suggested organizing the result sections as: Performance measurements of cockerels, Welfare measurements of cockerels, and Performance measurements of cockerels and pullets (or similar to avoid confusion). For clarity, indicate in Figures and Tables that data until 10 weeks of age refers to male and female chickens and that hens were relocated and cockerel pens were depopulated to 100 cockerels in each at 10 weeks of age. As done in Table 3 (Line 246-247). Otherwise it can be misleading for readers.

Figure 1: Legend is missing in.

Figure 1 and Tables 1 & 2 were made from the same dataset and do not add additional information. I suggest removing Table 1 and 2 because the Figure 1 captures this information already. Also, the main result in Table 2 is described in Lines 190-193.

LINE 208-214: The significance pairwise comparison between focal and random cockerels for the same genotype and age can be illustrated in Figure 1 with starts.

In Table 4, indicate the highest possible score of each term (e.g. as a footnote) and the period or age at which these data was collected.

Line 265: Indicate the average score for cleanliness and breast feather coverage among cockerel genotypes.

LINE 359-360: This result statement refers to data from both sexes.

Author Response

Dear Reviewer #2,

We thank you for your valuable comments and suggestions. In the attached file, we report any changes to the text and additional explanations. We are positive that we improved the manuscript accordingly and hope that these alterations find your support.

Yours sincerely,

the authors.

Reviewer 3 Report

Manuscript ID: animals-915406

Title: Are dual-purpose chickens twice as good? Measuring performance and animal welfare throughout the fattening period

General comment;

The manuscript is adequately written, and focuses on an important and relevant area; the culling of day-old layer male chicks. However, the introduction, discussion and conclusion needs work, and the authors need to make the point clearer as to how dual-purpose chickens could ever be the alternative to chick culling. The authors state several times in the manuscript that dual-purpose birds are less efficient than pure breeder- and broiler lines, and will likely never replace the pure lines. Thus, dual-purpose birds may in the best case reduce the number of culled chicks. The aim of the study needs to be clear throughout the manuscript, and the intro, discussion and conclusion must focus on the aim and the results of the study.   

Specific comments:

L13: rephrase “alternatives for the culling” – this paper does not provide alternatives to culling

L23: again, “three alternatives to culling” should be rephrased, you investigated the performance of these three birds

L31: does the FCR and EPEF argue for an earlier slaughter date in all birds, or for the LD only? “date” should be replaced with “age”

L31-32: conclusions should focus on the results, be more precise. In what way could focus on FCR underpin the necessity of further research? In order to improve the performance of the breeds?   

L44: the authors use different terms for the male chicks (cocks, male chicks, cockerels, brother cocks). Use one common term throughout.

L53-54: The meaning of these sentences are unclear, what is the authors aiming at, what are the differences between weighing random birds and sampling methods?

L55-56: this statement need a reference or elaboration, with hundreds of published papers on broiler growth, has there ever been a discussion to sampling methods?  

L57: replace “address” with “includes”

L58: insert “animal” before “welfare”

L58: replace “these” with “the”

L58-59: the reference is referring to cattle, the authors need references to this being true for poultry, or modify sentence.

L61: this reference is to learning, not to better adaption by native genotypes, which the sentence implies.

L61-62: again, these references do not support the claim that native breeds are better adapted to a production systems, that this paragraph infer. Either include other references or rephrase sentence.

L70-74: the WQ protocol is also aimed at laying hens, which spend more time in the farm, does the authors imply that the WQ is not appropriate for laying hens? If the point of the paragraph is that this study performs the WQ weekly, and thus provides a more reliable welfare assessment, this should be made more clear.

L77: “fattening performance”, specify this term previously in the intro

L78: “behavioural development”, the only behaviour included in the study is the QBA, this should be specified

L103-104: what was the reason that all birds were fed according to conventional fattening of broilers? Are there not specific recommendations for the Lohman birds for instance? Or was the feed a planned study design?  

L159-160: This is not a common way to present QBA data, include a reference to this method. Why did you not use a PCA analysis?  

L185: replace “compare” with “see”

L195-197: “differed from each other” – in what regard? Clarify these sentences

L200: Figure 1: include legends to genotypes and focal/random groups in figure

L216: remove “,” after LD

L226: First sentence can be deleted, or rephrased.

L235: The sentence “Genotypes differed in their FCR..” is not clear, specify results to genotypes.

L250: Table 3: maybe include average FCR per genotype at the end of table?

L281-282: What is meant here? How are the “welfare concerns seen uncritically?”

L281: generally, for the discussion, stay to the aims described in the intro, your aims were 1)

investigate the fattening performance 2) focal/random and 3) health and behaviour. The discussion should keep to the same structure

L281-284: this paragraph makes it seem as though the aim of the paper was to evaluate the welfare of these males throughout the growing period.

L283-285: QBA studies on poultry, and broilers specifically should be referred here, e.g. Muri et al., 2019 on QBA in broilers.

L288: specify “scores” – do you mean QBA scores?

L289-292: this is unclear. If RL showed lower values in a range of different expressions, like playful, active and fearful, how can you make the statement in L291 that RL showed “reduced positive emotional state”?

L292: replace “hybrid lines” with LB and LD

L292-294: This is not a well-constructed sentence. What is meant with “although they have not been selected for”? and how were they less active? By means of the QBA?

L294-296: again, an unclear sentence. What is meant here, should the breeding select for more or less active birds? And why? And what is “behavioural welfare”?

 L358: I would spend a paragraph in the discussion on the extreme low mortality of the LB birds (0.75%). Sustainability could also be affected positively by reduced mortality?

L380-381: this does not belong in the conclusion, you did not find an alternative to culling of day-olds. The conclusion is for summarizing of the main results, not to continue the discussion. Please rephrase the conclusion and focus on your results (L380-391)

Author Response

Dear Reviewer #3,

We thank you for your valuable comments and suggestions. In the attached file, we report any changes to the text and additional explanations. We are positive that we improved the manuscript accordingly and hope that these alterations find your support.

Yours sincerely,

the authors.

Round 2

Reviewer 1 Report

Genotype is not the correct term to be used. Genotype refers to the genetic makeup of an individual. You are not measuring individual animal genetic codes. Genetic line or strain would be more appropriate. Change throughout.

Simple summary: The simple summary should be written in lay terms so that members of the general public could understand the background of the research, the methodology, and results. In general, I do not think the simple summary does this.

Abstract: The abstract does a poor job of explaining the purpose of the study and the details of the study to allow a reader to understand the study on its own. What is the background of this project and why is it important? What was the sample size? How long were the birds observed? You also do not provide any statistical results in the abstract.

L32-33: 9 weeks, 13 weeks, 15 weeks... of what? 15 weeks of age?

L33: “The FCR and EPEF argue for a slaughtering age around week 8...” A better way to say this would be “The results of the FCR and EPEF suggest that cockerels should be slaughtered around...”

L34-35: I’m not sure what you mean by this sentence.

L36-37: Were the birds in an intensive or extensive production system? Based on the description in the methods, I would guess intensive.

L37-39: The last sentence is vague and could be improved.

Introduction: I still feel like the introduction could be better organized to help the reader understand the purpose, significance, and aims of the research.

L48: What are male layer hybrids? Briefly explain the differences between birds used for different purposed to set up why you used the strains you did and help the reader understand the problem and the potential solutions.

L57: Why is this an important question?

L59-64: What does this have to do with the purpose and aims of the study?

L104: What about your hypothesis for the focal vs. random sampling?

L104: How is the native breed more adapted to extensive rearing? What do you expect to find for behavior and welfare?

Materials and Methods:

L118: Three pens? How many birds per pen?

L153: Do you mean all the birds were kept within one pen/room?

L182: Why was inquisitive excluded?

L185-186: Was it always the same familiar person?

Results

L287: What do you mean by location of the hatch?

L324-325: I’m not sure what this means. 20% of the RL birds had slight swelling, but not corresponding to any other scale value? Does this mean they had foot pad dermatitis, but feathers and hocks were good?

Discussion:

L351-358: You do not mention the goal of comparing focal and random birds.

L368: Do you have a source to back up this claim?

L376-377: What does genotype in process mean?

L453-454: The results do not mention that RL have lower QBA score.

Reviewer 3 Report

I have no further comments.